# Study protocol for a randomized controlled trial: Integrating the 'Time-limited Trial' in the emergency department

Tadayuki Hashimoto[1,2]* *, Rachel K. Putman[3], Anthony F. Massaro[3], Youkie Shiozawa[1], Katherine McGough[4], Kerry K. McCabe[5], Judith A. Linden[5], Wei Wang[6], Shan W. Liu[7,8], Maura Kennedy[7,8], Thanh H. Neville[9], Jacqueline M. Kruser[10], Rebecca L. Sudore[11], Mara A. Schonberg[7,12], James A. Tulsky[13,14], Kei Ouchi[1,7,13,15]*

1 Department of Emergency Medicine, Brigham and Women's Hospital, Boston, Massachusetts, United States of America, 2 Department of General Medicine, Osaka Medical and Pharmaceutical University, Takatsuki, Osaka, Japan, 3 Department of Pulmonary and Critical Care, Brigham and Women's Hospital, Boston, Massachusetts, United States of America, 4 University of Missouri School of Medicine, Columbia, Missouri, United States of America, 5 Department of Emergency Medicine, Boston Medical Center, Boston, Massachusetts, United States of America, 6 Survey and Data Management Core, Dana-Farber Cancer Institute, Boston, Massachusetts, United States of America, 7 Department of Emergency Medicine, Harvard Medical School, Boston, Massachusetts, United States of America, 8 Department of Emergency Medicine, Massachusetts General Hospital, Boston, Massachusetts, United States of America, 9 Division of Pulmonary and Critical Care, Department of Medicine, David Geffen School of Medicine, UCLA, Los Angeles, California, United States of America, 10 Division of Pulmonary and Critical Care Medicine, Department of Medicine, University of Wisconsin, Madison, Wisconsin, United States of America, 11 Division of Geriatrics, Department of Medicine, University of California, San Francisco, United States of America, 12 Department of Medicine, Beth Israel Deaconess Medical Center, Boston, Massachusetts, United States of America, 13 Department of Psychosocial Oncology and Palliative Care, Dana-Farber Cancer Institute, Boston, Massachusetts, United States of America, 14 Division of Palliative Medicine, Department of Medicine, Brigham and Women's Hospital, Boston, Massachusetts, United States of America, 15 Serious Illness Care Program, Ariadne Labs, Boston, Massachusetts, United States of America

☺ These authors contributed equally to this work.
* thashimoto2@bwh.harvard.edu

## Abstract

### Introduction

Time-limited trial (TLT) is a structured approach between clinicians and seriously ill patients or their surrogates to discuss patients' values and preferences, prognosis, and shared decision-making to use specific therapies for a prespecified period of time in the face of prognostic uncertainty. Some evidence exists that this approach may lead to more patient-centered care in the intensive care unit; however, it has never been evaluated in the emergency department (ED). The study protocol aims to assess the feasibility and acceptability of TLTs initiated in the ED.

### Methods and analysis

We will conduct a parallel group, clinician-level, pilot randomized clinical trial among 40 ED clinicians. We will measure feasibility (e.g., the time it takes to conduct the TLTs by ED clinicians) and clinician and patient-reported acceptability of the TLT, and also track patients' clinical outcomes via medical record review.

**Data Availability Statement:** Deidentified research data will be made publicly available when the study is completed and published.

**Funding:** The author(s) received no specific funding for this work.

**Competing interests:** Dr. Kei Ouchi has received funding personally from Jolly Good, Inc (a virtual reality company) for consulting. This does not alter our adherence to PLOS ONE policies on sharing data and materials. There are no patents, products in development or marketed products associated with this research to declare.

## Discussion

This study protocol will investigate the potential of TLT initiated in the ED to lead to patient-centered intensive care utilization. By doing so, the study intends to improve palliative care integration for seriously ill older adults in the ED and intensive care unit.

## Trial identifier and registry name

ClinicalTrials.gov ID: NCT06378151 https://clinicaltrials.gov/study/NCT06378151; Pre-results; a randomized controlled trial: Time-limited Trials in the Emergency Department.

## Introduction

In the United States (U.S.), approximately 87 older adults ($\geq$ 65 years) require intensive care in emergency departments (ED) every hour [1]. This often occurs without these patients understanding the high risk of acquiring new disability [2]. During the last six months of life, 75% of older adults visit the ED [3], and intensive care is started in 30% of these patients [4]. Intensive care use in this population increased from 18.5% in 1993 to 24.7% in 2002 [5], resulting in an annual healthcare expenditure of $108 billion (2010, 4.1% of U.S. national healthcare expenditure) [6]. Shared decision-making around goals of care plays a key role in this population, with more than 70% of older adults reporting that at the end of life, they prefer to focus on quality of life rather than life extension [7] and would even give up one year of life to avoid dying in the intensive care unit (ICU) [8]. Despite a strong preference against invasive interventions, a systematic review revealed that 56% to 99% of adults do not have advance directives at the time of an ED visit [9]. Since the decision to initiate intensive care often occurs in the ED, ED clinicians recognize the opportunity to provide more patients-centered care through shared decision making [10]. However, in the time-pressured ED environment, no standardized methods exist to guide ED clinicians in leading critical shared decision-making conversations regarding initiating intensive care [11–13].

Experts agree that serious illness care should focus on preparing people for and supporting them in in-the-moment decision-making during clinical deterioration [14–16]. Such medical crises exemplify the most challenging, in-the-moment decision-making. In situations where there is prognostic uncertainty. **Time-Limited Trial (TLT)** is a potential approach to reducing potentially nonbeneficial treatments for seriously ill older adults [17–20]. TLT is structured approach between clinicians and patients/surrogates to discuss patients' values and preferences, prognosis, and shared decision-making to use specific therapies for a prespecified period of time. TLT allows both clinicians and patients/surrogates to regularly assess if therapies are meeting patients' goals over structured time intervals. Guided by the Capacity, Opportunity, Motivation to perform a Behavior theoretical framework, which provides a lens to understand and guide behavior change in healthcare setting [21], a quality improvement study of TLTs in the ICU (N = 209) was associated with a 35% increase in formal family meetings (p<0.01), 45% increase in clinicians eliciting values and preferences of patients (p<0.01), 1.3 day decrease in median ICU length of stay (p<0.02), and 13% decrease in mechanical ventilation (p = 0.02). No change in hospital mortality was observed (58.4% vs. 58.3%, p = 0.99) [22, 23].

In 2023, the American Thoracic Society published an official workshop report authored by more than 100 stakeholders. It defined TLT in critical care as: a collaborative plan among

clinicians and a patient and/or their surrogate decision-maker (s) to use life-sustaining therapy for a defined duration, after which the patient's response to therapy informs the decision to continue care directed towards recovery, transition to care focused exclusively on comfort, or extend the trial's duration" [20]. TLT has been also published as an expert-recommended approach in Annals of Emergency Medicine [24]. Decisions to initiate intensive care often occur in ED, yet no study has evaluated the effect of TLTs when initiated in this setting. Whether initiating TLTs earlier, in the ED, is feasible remains unknown.

In the ED, emergency clinicians are responsible for the initial assessment and treatment of critical conditions. Similar to other clinical initiatives that have a lasting impact on patients' care when initiated in the ED (e.g., the first dose of antibiotics for sepsis, thrombolysis for acute ischemic stroke, etc.), for seriously ill older adults being admitted to the intensive care unit, emergency clinicians' role is to prepare patients/surrogates for the anticipated difficult clinical course, help formulate and express their values and goals, and set the expectation to re-discuss these goals based on the clinical progression of the patient over a defined period of time. Given the limitations of the ED setting with clinical uncertainty about prognosis and anticipated outcomes, we used a comprehensive approach to refine the TLT conversation guide for use in this setting.

We anticipate that systematic application of the TLT adapted for use in the ED will lead to more patient-centered utilization of intensive care for seriously ill older adults. Implementing TLTs from the ED is expected to encounter numerous barriers; thus, our initial study aims to assess their feasibility and acceptability.

## Objectives

The objective is to assess the feasibility (Aim 1) and the acceptability to clinicians and patients (Aim 2) of TLTs initiated in the ED. Our central hypothesis is that, in randomized study design conducted in the ED, TLTs will be feasible and acceptable.

**Aim 1: Feasibility.**  We will determine the feasibility of conducting a clinician-level, randomized controlled trial of TLTs compared to usual care in the ED, hypothesizing that TLTs are feasible to conduct in a clinician-level randomized study in the ED.

**Aim 2: Patient-reported acceptability.**  We will determine the patient-reported acceptability of TLTs in the ED, hypothesizing that the patients or surrogates (if patients are unable to consent) randomized to TLTs will feel more often heard and understood about their end-of-life care wishes (primary outcome).

## Materials and methods

### Trial design and setting

**Overview.**  We will conduct a parallel group, clinician-level, pilot randomized clinical trial among 40 ED clinicians (~20 in the TLT arm and ~20 in usual care). Fig 1 shows the Standard Protocol Items: Recommendations for Interventional Trials (SPIRIT) 2013 schedule of enrollment, interventions, and assessments. Fig 2 includes a flow chart of the main parts of the study methods.

**Setting.**  This study will be conducted in an urban, academic ED with 747 hospital and 110 Intensive Care Unit (ICU) beds and a suburban, community ED with 171 hospital and 14 ICU beds. Approximate annual ED visits are 60,000 and 30,000, respectively.

**Randomization.**  Using REDCap's integrated randomization module [25], we will conduct clinician-level randomization using random blocks of four to ensure that ED clinicians are evenly distributed one-to-one between intervention and control arms.

| | | STUDY PERIOD | | | |
|---|---|---|---|---|---|
| | | Enrolment | Allocation | Post-allocation | Close-out |
| TIMEPOINT | | *Baseline* | *1st Month* | *3 Months* | *6 Months* |
| ENROLMENT | | | | | |
| | Eligibility screen | X | | | |
| | Informend consent | X | | | |
| | Allocation | | X | | |
| INTERVENTIONS | | | | | |
| | Experimental group intervention | | X | | |
| | Regular control group intervention | | | | |
| ASSESSMENTS | | | | | |
| Clinician outcome variables | Experimental group intervention | | | ←————————————→ | |
| | Regular control group intervention | | | | |
| Patient outcome variables | Experimental group intervention | | | ←————————————→ | |
| | Regular control group intervention | | | ←————————————→ | |

**Fig 1. SPIRIT 2013 schedule.** SPIRIT schedule of enrollment, interventions, and assessments timepoints along the study period.

**Aim 1: Feasibility.** We will conduct the randomized trial and measure the feasibility outcomes (e.g., the time it takes to conduct discussion about TLTs).

**Aim 2: Patient-reported acceptability.** Among the patients who will be cared for by the ED clinicians enrolled in the randomized trial, we will measure the patient-reported acceptability of TLTs (intervention arm only) and track their clinical outcomes (e.g., Number of days to the first family meeting in ICU, ICU length of stay, see Outcomes below) via electronic health record (EHR) review (both intervention and control arms). If, for some reason, such as full capacity, a patient who meets the criteria for ICU admission is admitted to a general ward instead, their data will also be checked if TLTs have been performed in the ED.

## Inclusion and exclusion criteria

**Aim 1: Feasibility.** ED clinicians are subjects given they will answer questions (i.e., clinician-reported feasibility/acceptability questionnaire) after conducting TLTs in the ED. ED clinicians will be randomized to intervention (TLT training) vs. usual care (no TLT training).

*Inclusion.* Attending clinicians, resident clinicians, or mid-level providers working in the ED who are willing to be randomized to become the study clinicians assigned as intervention group.

*Exclusion.* ED clinicians unwilling to consent and be randomized to intervention TLT training.

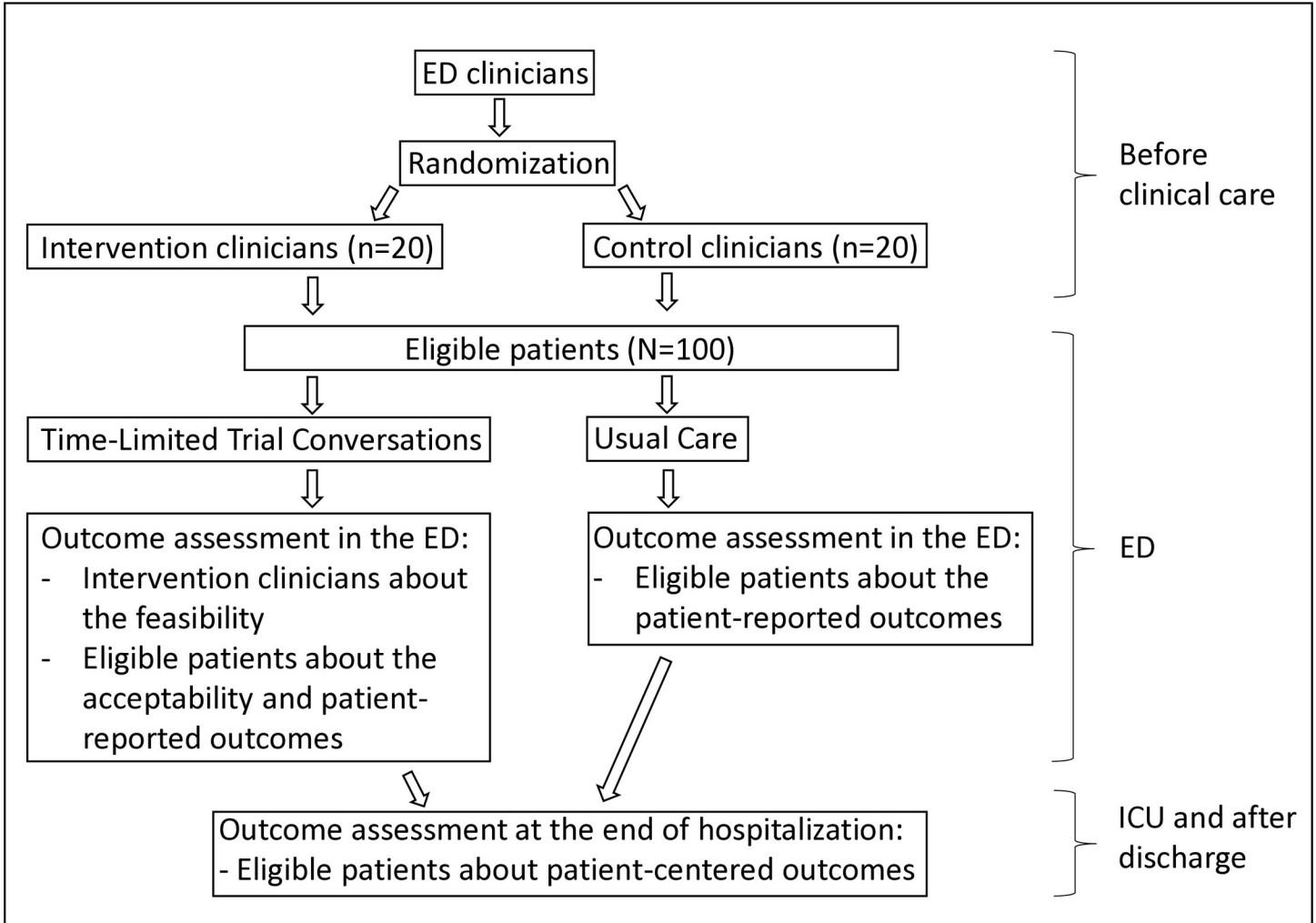

**Fig 2. Time-Limited Trials in the Emergency Department (ED): A study schema.** ED clinicians are divided into an intervention group and a control group, each consisting of 20 members. The intervention group is equipped with resources to implement TLTs. This group will identify and apply TLTs to suitable patients among those they treat. Outcomes will be measured for both clinicians and patients within these groups, with assessments of patients conducted in both the ED and the ICU.

*Recruitment and consensus.* With the Department Chair's approval and after reviewing the ED roster, we will announce the study in staff meetings and solicit participation. After introducing the study at staff meetings, the study team will obtain verbal, informed consent with a study information sheet.

**Aim 2: Patient-reported acceptability.** The subjects will be seriously ill older adults, or their surrogates being cared for by the participating ED clinicians (both the intervention or control arms–see details on which outcome to be assessed for which arm in *6 Study Procedures*, *Outcome Assessment* below). Being seriously ill here refers to the applicable condition in the Inclusion criteria. If the ED clinicians determine that the patient is not able to provide consent due to cognitive impairment, dementia, delirium, or critical illness, the surrogates will be asked to participate in TLTs (Table 1).

*Inclusion.*

a. ≥50 years or older with ≥one serious life-limiting illness* being admitted to the intensive care unit in the ED; or

**Table 1. Intervention target and outcome assessment.**

| Ability to consent | Intervention target | Outcome assessment |
|---|---|---|
| Present | Patient | Patient-reported acceptability + EHR clinical outcomes of the patient |
| Absent | Surrogate | Surrogate-reported acceptability + EHR clinical outcomes of the patient |

b. ≥75 years or older being admitted to the intensive care unit in the ED; or

c. ED clinicians will not be surprised if the patient dies in the current hospital admission (i.e.; deem patients to be at high risk of death/disability from the current disease process); and

d. English speaking

*Serious illness criteria with high one-year mortality are selected based on best practice recommendations [26] similar to prior studies [27–29]: 1) stage III/IV or metastatic cancer [30, 31]; 2) end-stage renal disease on dialysis [32]; 3) chronic heart/lung disease requiring home oxygen supplementation or experiencing shortness of breath with walking [33, 34]; 4) moderate to severe dementia (surrogate required for enrollment); or 5) ≥2 hospitalizations in the past six months [30, 35, 36].
*Exclusion.*

a. Unable or unwilling to provide informed consent; or

b. Non-English speaking; or

c. Clinically inappropriate, determined by ED clinicians, and no surrogate is available

## Recruitment methods

We will identify potentially eligible patients using the ED track board in the EHR. Upon completion of TLTs by the assigned ED clinician (who also provides clinical care for the patient), our study team (e.g., research assistant, PI) will obtain verbal, informed consent from the patient or surrogate (if applicable) in the ED. The capacity to consent the patient or surrogate is determined by the assigned ED clinician who conducted TLTs.

Regarding the recruitment period, the recruitment of clinicians will commence in anticipated May 2024 and will conclude after three months. Following this, the recruitment of patients is scheduled to begin in August 2024, with the aim of collecting the necessary sample size over the course of half year. The recruitment of patients is scheduled to be completed by no later than February 2025.

## Intervention

**Experimental group.** Clinician-led conversations about patient's values and goals, potential benefits and risks of intensive care, and anticipated outcomes are standard of care in the ED. TLT is a multi-modal intervention that provides patient-centered structures to this standard of care. In this evaluation of TLTs in the ED, the intervention components will include: 1) a structured conversation guide; 2) clinician training to use the guide; 3) an EHR template for documentation; and 4) standardized communication with the intensivist (Fig 3).

1. The TLT Conversation Guide: The structured conversation guide entails discussing patients' values and goals, prognosis, and shared decision-making to use a trial of intensive care. Originally used in ICU settings [22, 23], we systematically refined the guide to

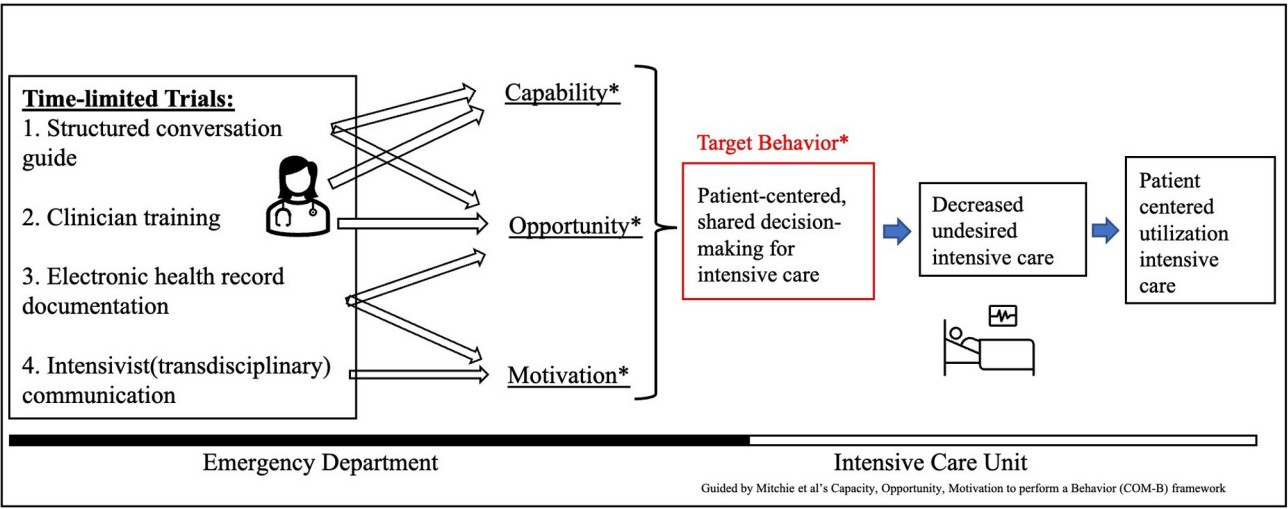

**Fig 3. Conceptual model for Time-Limited Trials in the emergency department.**

be used in the ED incorporating inputs from patient advisors' and ED clinicians' inputs (S1 Appendix). Guided by Stirman's framework for adapting efficacious interventions for new populations [37], we assessed the content validity of the original ICU TLT guide for use in the ED to seamlessly allow consistent care expectations and transition to the ICU team. We evaluated the following domains: necessity, clarity, relevance to the ED, and comprehensiveness. We involved three intensivists, five emergency clinicians, and three patient advisors to review and provide feedback on the refined intervention script and study approach. They rated the TLT conversation guide on a 4-point scale (e.g., 1 = irrelevant/not clear, 4 = essential/very clear) based on each section's (1) necessity, (2) clinical relevance, and (3) clarity in the ED settings. The ratings were then used to compute a content validity index (CVI; 0–1.0, ≥0.7 is considered adequate [38]). Specifically, this expert panel identified that the following aspects are uncertain in the ED: prognosis (and sometimes diagnosis), indicators for clinical improvement, and timeframe for the TLT. Therefore, we iteratively refined these three points until they were deemed acceptable for use by emergency clinicians. As a result, the refined guide intentionally communicates the uncertainty and clinician worries simultaneously (e.g., "Based on what we know so far, our hope is that [patient] improves with these treatments. Ath the same time, I also worry the treatments may not reverse the problem.").

2. Clinician Training: Clinician training will include a one-hour didactic on research methodologies and serious illness communication skills, followed by a four-hour communication training with trained actors. Similar training has been described previously [39–41].

3. EHR Documentation: An EHR template for documenting the findings of TLTs has been developed.

4. Intensivist Communication: A standard template to communicate findings of TLTs to the intensivists has been developed.

**Control group.**   ED clinicians assigned to the control group will not receive the TLT training/guide. Patients cared for by the control group ED clinicians will receive usual care in the ED. No restrictions will be placed on serious illness conversations in the control group.

## Data collection

We will collect the following variables as primary and secondary outcomes during and after the TLT in the ED (Table 2).

For Aim 1 Feasibility, we will collect all outcomes from the intervention arm, and EHR outcomes from both intervention and control arms. Descriptive statistics will be used for baseline characteristics of the study sample. We will define feasibility to be the following:

- The median time to complete the introduction of TLTs is <10 minutes (primary outcome).

- >70% of intervention components are completed, assessed using an intervention fidelity checklist by an observer/research team member (S2 Appendix).

- >70% of clinicians in the intervention group reporting "somewhat feasible (4)" or "completely feasible (5).

- We will track recruitment, participation, and retention rates of assigned clinicians, including reasons for not participating. We will track whether inpatient clinicians authored the serious

**Table 2. Outcomes and assessment timing.**

| Aim 1: Feasibility (* indicates outcomes collected only from the intervention arm) | | |
|---|---|---|
| **Outcomes** | **Descriptions** | **Timing** |
| Time to complete the introduction of TLTs (primary outcome) * | With direct observation, the study team will record how long it takes to complete the introduction of TLTs by the ED clinician assigned as intervention group. | At the time of the introduction of TLTs |
| Intervention fidelity* | With direct observation, the study team will record the completion of TLTs components using an intervention fidelity checklist (S2 Appendix). | At the time of the TLT conversation |
| Clinician-reported feasibility* | The study team will ask the assigned ED clinician to complete two 5-point Likert scale questions (i.e., "How feasible was it to conduct the introduction of TLTs with your patient?" and "How likely would you recommend the TLT conversation for your colleagues?" "Not at all (1)" to "Completely (5)"). | Immediately after the introduction of TLTs |
| Recruitment* | The percent that agrees to conduct the introduction of TLTs | At enrollment |
| Clinician-reported satisfaction* | The study team will ask the assigned ED clinician to complete 5-point Likert scale questions (i.e., "How satisfied are you with the conversation? "Not at all (1)" to "Completely (5)"). | Immediately after the introduction of TLTs |
| EHR documentation by inpatient clinicians (i.e., intensivists) | Review EHR for new documentation of reference to TLTs, serious illness conversation, change in code status, or advance directive forms by inpatient clinicians. | After 24 hours, 48 hours, and 1 week |
| Aim 2: Patient-Reported Acceptability (* indicates outcomes collected only from the intervention arm, ** indicates outcomes collected when a patient who is eligible for ICU admission is admitted to a general ward) | | |
| **Outcomes** | **Descriptions** | **Timing** |
| Patient-reported acceptability of the introduction of TLTs (primary outcome) * | A 5-point Likert scale (i.e., "How acceptable was it for your doctor to talk to you about your expectations for ICU care?" and "How likely would you recommend this conversation for other patients like you?" "Not at all (1)" to "Completely (5)"). | Immediately after the introduction of TLTs |
| Heard and understood | A National Quality Forum endorsed, validated measure for palliative care modified to fit the context of serious illness conversations ("How well they feel heard and understood by your emergency department clinician about the medical care they would want if they were to get sicker?" a 5-point Likert scale "Not at all (1)" to "Completely (5)") [44]. | Immediately after the introduction of TLTs |
| Patient-reported end-of-life quality of communication | A validated, quality of end-of-life communication survey (4 items selected a priori, a 10-point Likert scale "Worst you can imagine (0)" to "Best you can imagine (10)") [45]. | Immediately after the the the introduction of TLTs |
| Decisional regret scale | A validated survey that measures "distress or remorse after a [health care] decision" (5 items, 5-point Likert scale "Strongly agree (1)" to "Strongly disagree (5)") [46]. | After 24 hours, 48 hours, and 1 week |
| Clinical outcomes abstracted from EHR | Number of days to the first family meeting in ICU, ICU length of stay (LOS), hospital LOS, Number of family meetings, ICU procedures (e.g., CPR, pressors, etc.), ICU mortality, hospital disposition, hospice utilization.<br>Number of days to the first family meeting in ward**, number of family meeting in ward**, number of goals of care discussion in ward **, number of days and timing of clinical documentation of ACP in ward ** | After discharge or upon patient death |

illness conversations in EHR. The EHR-based outcome will be assessed using an EHR review. We will publish our findings regardless of the results and will follow the CONSORT guidelines for reporting RCT [42].

For Aim 2 Patient-reported acceptability, we will collect data from both the intervention and control arms. We predict that the patients and surrogates undergoing the introduction of TLTs will feel heard and understood about their end-of-life care wishes (primary acceptability outcome). Descriptive statistics will be used for baseline characteristics of the study sample. We will compare patient demographics between study arms to assess randomization using two sample t-test or Wilcoxon rank sum test for continuous variables and chi-square for categorical variables. Within arms, we will use a one-sample binomial exact test of proportions for categorical outcomes (e.g., EHR documentation), and Wilcoxon signed ranks test for ordinal outcomes (e.g., Heard and Understood survey). We will conduct a secondary analysis using linear mixed models at baseline and one month to include all non-missing data. The EHR-based outcome will be assessed using an EHR review. We will publish our findings regardless of the results and will follow the CONSORT guidelines for reporting RCT [42].

The current study is in Phase I of intervention development. As such, the primary purpose of this study is to demonstrate the feasibility of implementing the intervention and randomizing clinicians to become interventionists. We used the standardized framework for feasibility studies previously described [43]. The feasibility is defined as:

1. Time to complete the introduction of TLTs (primary outcome): < 15 minutes is required to complete the introduction of TLT conversation by emergency clinicians.

2. Intervention fidelity: > 75% of clinicians complete more than 75% items on TLT Fidelity checklist.

3. Clinician-reported feasibility: > 3.5 is reported by physicians on average.

4. Recruitment: > 50% of eligible and willing patients are enrolled.

5. Clinician-reported satisfaction: > 3.5 is reported by physicians on average.

6. EHR documentation by inpatient clinicians (i.e., intensivists): > 75% of the medical records of patients who received the intervention include EHR documentation related to TLT.

In addition, we seek to follow the clinical outcomes of patients under the care of our interventionist clinicians to explore the potential effect of such clinician intervention. At this stage, we needed to balance the practicality of recruiting emergency clinicians to train in the TLT conversation guide (a substantial effort for busy clinicians) and the scientific aim of demonstrating the feasibility of such a study while exploring the clinical effects (e.g., How long does it take for the interventionist clinicians to care for enough seriously ill older adults to glean at the clinical effect?). Therefore, our interdisciplinary team will recruit 20 intervention and 20 control clinicians (40 total) and follow the clinical outcomes of their patients (N = 100).

## Data management

Our team will extract data from the EHR using a standardized approach [47]. All study data will be collected and stored using REDCap. A research assistant blinded to treatment allocation will perform all follow-up assessments.

The database system provides for secure web-based data entry with the data stored on servers maintained by Mass General Brigham. The data is encrypted during transmission. Consistent with NIH guidelines, a Data Safety Monitoring Committee will not be required for this

single site, interventional trial. We do not anticipate sharing the data with collaborators outside Mass General Brigham.

## Statistical analysis

No previous two-arm trial of TLT have been conducted; thus, the effect size remains unknown. Given the primary purpose of conducting the RCT is to demonstrate the feasibility, no gold standard for sample size calculations exists. However, prior literature suggests that 15 to 20 participants per group are required to ensure the scientific validity of the pilot study results [48]. Therefore, the present study will include a total of 40 participants with 20 participants per group. While pilot studies cannot provide adequate power to determine the effect size for planning future studies, we intend to explore the potential clinical signal of the TLT intervention. We plan to report descriptive statistics as follows in order to estimate these as accurately as possible (Table 3).

## Ethics

Our project was approved by the Institutional Review Board at the Brigham and Women's Hospital (project number: 2023p002243). In addition, this study protocol has a trial identifier and registry name at ClinicalTrials.gov, ID: NCT06378151 Written, informed consent will be obtained from all participants involved in the study and can withdraw consent at any time.

**Adverse Events (AE) and Unanticipated Problems (UP).** Any AE will be reported to our institutional review board at the time of the annual continuing review by the principal investigator using an AE form in accordance with the Mass General Brigham Human Research Committee guidelines for AE reporting. As with any AE above, the study team will report any UP within 5 working days / 7 calendar days of the date. The investigator first becomes aware of the problem using an Adverse Event Form in accordance with the Mass General Brigham Human Research Committee guidelines for AE reporting following our institutional policy.

**Study termination.** We will follow the participating clinicians for up to one year to accrue an adequate number of patients they admit to the intensive care unit. Given that an ED clinician may encounter ~3 to 10 seriously ill older adults being admitted to the intensive care unit per month, we estimate that it would take approximately one year to accrue ~10 patient/ assigned ED clinicians who might meet the inclusion/exclusion criteria for Aim 2. The follow-

**Table 3. Outcomes and descriptive statistics.**

| Outcomes | Variable Type | Descriptive Statistics |
|---|---|---|
| Time to complete the introduction of TLTs (primary outcome) | Continuous | Mean and Standard Deviation (if normally distributed); Median and Interquartile Range (if not normally distributed) |
| Intervention fidelity | Categorical | Frequency (n) and Percentage (%) |
| Clinician-reported feasibility | Continuous | Mean and Standard Deviation (if normally distributed); Median and Interquartile Range (if not normally distributed) |
| Recruitment | Categorical | Frequency (n) and Percentage (%) |
| Clinician-reported satisfaction | . Continuous | Mean and Standard Deviation (if normally distributed); Median and Interquartile Range (if not normally distributed) |
| EHR documentation by inpatient clinicians | Categorical | Frequency (n) and Percentage (%) |

up assessments of the patients will terminate when/if the participants die or are discharged from the hospital.

**Remuneration.** For Aim 1 Feasibility, we will compensate $500 to clinicians randomized to the intervention arm for one-hour didactic and four-hour communication training. In addition, we will compensate $25 for every patient that the introduction of TLT is completed. For control arm clinicians, we will compensate $100 after randomization. For Aim 2 Patient-reported Acceptability, we will compensate patients $20 for completing the surveys.

## Limitations

This study recognizes several limitations. One significant issue affecting internal validity is the variability among clinicians and patients. There is a chance of variability in how interventions are delivered because ED clinicians with diverse clinical experience levels will be recruited. The risk of intervention consistency is reduced by employing a structured TLT conversation guide, providing serious illness communication training based on evidence, and using a check-list to ensure fidelity to the intervention. Moreover, any differences due to clinicians' previous experiences are neutralized through random assignment at the clinician level. Variability in the patient population seen by intervention clinicians is also a possibility. Efforts will be made to include a diverse patient group by selecting clinicians with varied work schedules, including different shift patterns. In the realm of emergency medicine, neither intervention nor control clinicians can choose their patients, thereby ensuring random selection.

Additional variation exists in clinician-patient communication within the ED. To counter-act this, the study will introduce a structured conversation guide, provide training in commu-nication grounded in evidence, and supervise conversations with an experienced clinician. On the patient front, the wide range of backgrounds among emergency patients offers a challenge, which is addressed by randomizing clinician selection and standardizing their shifts to pro-mote a randomized patient assignment.

Clinicians participating in the study will receive payment for completing the introduction of TLTs. Although this may differ from typical clinical practice, the payment aligns with what insurance provides for Advance Care Planning (CPT codes 99497 and 99498) [49].

Implementing serious illness communication training for all ED clinicians might be chal-lenging in real-world clinical contexts. However, research by Grudzen et al. has shown that it is feasible to spread this training widely across 35 ED sites using virtual methods [50]. Further-more, the increase in both the number of trainees and trainers in serious illness communica-tion training indicates a growing trend in this field [51].

## Discussion

We describe an innovative protocol for a randomized trial to assess the feasibility and accept-ability of initiating TLT in the ED. This study is important as the successful execution of this study heralds the standardization of one of the most challenging, in-the-moment conversa-tions for ED clinicians. Given the global incidence of these challenging conversations with aging populations worldwide, the potential exists to disseminate an evidence-based structure to these conversations. Specifically, the systematic integrating of TLTs in ED carries two significances.

Initially, the study highlights the patient-centered care in the ED. Current emergency care often falls short of providing patient-centered care to older patients [39, 52]. By integrating TLT into emergency care, there is a potential to significantly enhance patient-centeredness. This approach could lead to more personalized and appropriate care for older patients,

aligning treatment goals with individual preferences and needs, and improving overall patient satisfaction and outcomes in emergency care settings.

Secondary, it addresses the economic dimensions of healthcare. Existing research underscores that the initiation of TLTs in ICU is efficacious in curtailing ICU length of stay [22, 23]. The American Heart Association has been vocal about the necessity of ED and ICU collaboration in the survival continuum [53]. Presently, this collaborative effort is chiefly concentrated on intensive care for acutely ill patients, particularly those suffering from cardiac arrest. There is an emerging imperative to broaden this collaboration to include palliative care as well. This expansion is anticipated to further reduce unnecessary or disproportionate intensive care and ICU length of stay.

Moreover, the study accentuates the importance of patient-centered care within the ED. Contemporary emergency care frequently fails to meet the patient-centered needs of seriously ill elderly populations [39, 53]. The integration of TLT into emergency care procedures promises a substantial enhancement in patient-centeredness. Such an approach will deliver intensive care aligned to patients' values and goals in the face of serious illness. By doing so, this study will expand the scope of ED-based care from acute, disease-oriented care (e.g., gunshot wounds) to include patient-centered care (e.g., value-based, end-of-life care) for seriously ill older adults by integrating geriatrics and palliative medicine principles.

In summary, successfully integrating TLT in ED offers a valuable opportunity to improve both efficiency of healthcare and the economics. More importantly, it can significantly improve the quality of care for older patients, a group whose numbers are increasing worldwide.

## Supporting information

**S1 Checklist. SPIRIT 2013 checklist: Recommended items to address in a clinical trial protocol and related documents.**
(DOC)

**S1 File. Detailed protocol.**
(DOCX)

**S1 Appendix.**
(DOCX)

**S2 Appendix.**
(DOCX)

## Acknowledgments

The study is made possible by generous philanthropic support from Dr. Alice Mozley. Dr. Mozley's contribution as a philanthropist and a patient advocate in this study's design and execution is greatly appreciated.

## Author Contributions

**Conceptualization:** Rachel K. Putman, Anthony F. Massaro, Katherine McGough, Shan W. Liu, Thanh H. Neville, Jacqueline M. Kruser, Rebecca L. Sudore, James A. Tulsky, Kei Ouchi.

**Funding acquisition:** Kei Ouchi.

**Methodology:** Wei Wang, Thanh H. Neville, Mara A. Schonberg, Kei Ouchi.

**Project administration:** Youkie Shiozawa.

**Supervision:** James A. Tulsky, Kei Ouchi.

**Visualization:** Mara A. Schonberg.

**Writing – original draft:** Tadayuki Hashimoto, Katherine McGough.

**Writing – review & editing:** Rachel K. Putman, Anthony F. Massaro, Kerry K. McCabe, Judith A. Linden, Wei Wang, Shan W. Liu, Maura Kennedy, Thanh H. Neville, Jacqueline M. Kruser, Rebecca L. Sudore, Mara A. Schonberg, James A. Tulsky, Kei Ouchi.

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
