## [Decision Letter · Decision Letter 0]

1 Oct 2024

PONE-D-24-11345Study Protocol for a Randomized Controlled Trial: Integrating the ‘Time-limited Trial’ in the Emergency DepartmentPLOS ONE

Dear Dr. Hashimoto,

Thank you for submitting your manuscript to PLOS ONE. After careful consideration, we feel that it has merit but does not fully meet PLOS ONE’s publication criteria as it currently stands. Therefore, we invite you to submit a revised version of the manuscript that addresses the points raised during the review process.

Overall, the protocol for implementing time limited trials in the emergency department is interesting.  I commend you on exploring tools to improve communication with families in these difficult clinical situations.  The reviewers raised several important critiques which are highlighted below.  I ask you to pay particular attention to the following issues:

Reviewer 1 appropriately comments that initiating time limited trials in the ED may work against prognostic accuracy and whether it may create communication discordance with ICU physicians.  This critique relates to a larger question about this protocol, namely, what are the advantages and disadvantages of implementing time limited trials in the ED (compared to ICU)?  Having TLTs initiated by ICU physicians would have the advantage of communication continuity, perhaps better rapport as there will generally be more time for family meetings in the ICU, availability of ICU nurses and other multi-disciplinary services, among numerous other factors.  One can argue that ICU physicians may also have more experience with the trajectory of critical illness and may be able to provide more detailed information on prognosis/hospital course.  It would significantly aid the rationale for the protocol to be more clear about why initiating TLTs in EDs is an advantage (compared to initiating in ICU) and potential drawbacks to doing it this way.  In addition, I would consider incorporating ICU physicians as stakeholders in the acceptability assessments.  The ICU physicians will be continuing the additional communications and shared decision-making, so including them seems crucial.

Reviewer 2 comments on sample size and analytic plans are critical to address.  In addition, it would be helpful to have more details on how the numerous assessments will be conducted.  For example, will there be a member of the research team to collect data on family meetings daily both day and night shifts?  How will they know that a family meeting will be conducted by a trained ED physician?   Who will distribute and collect surveys that require follow up (Decision regret scale)? 

We look forward to receiving your revised manuscript.

Kind regards,

Dong Wook Chang

Academic Editor

PLOS ONE

“Dr. Ouchi is supported by the National Institute on Aging (K76AG064434)”

3. Thank you for stating the following in the Competing Interests

“Dr.Ouchi has received funding personally from Jolly Good, Inc (a virtual reality company) for consulting”

We note that one or more of the authors are employed by a commercial company: Jolly Good, Inc.

5. We note that the original protocol that you have uploaded as a Supporting Information file contains an institutional logo. As this logo is likely copyrighted, we ask that you please remove it from this file and upload an updated version upon resubmission.

**Comments to the Author**

1. Does the manuscript provide a valid rationale for the proposed study, with clearly identified and justified research questions?

Reviewer #1: Partly

Reviewer #2: Yes

2. Is the protocol technically sound and planned in a manner that will lead to a meaningful outcome and allow testing the stated hypotheses?

Reviewer #1: Partly

Reviewer #2: Partly

3. Is the methodology feasible and described in sufficient detail to allow the work to be replicable?

Reviewer #1: Yes

Reviewer #2: Yes

4. Have the authors described where all data underlying the findings will be made available when the study is complete?

Reviewer #1: Yes

Reviewer #2: Yes

5. Is the manuscript presented in an intelligible fashion and written in standard English?

Reviewer #1: Yes

Reviewer #2: Yes

6. Review Comments to the Author

You may also provide optional suggestions and comments to authors that they might find helpful in planning their study.

Reviewer #1: This is an interesting tool to evaluate but the overall goal of pursuing Time Limited Trials in the ED is not fully clear. The second sentence of the Discussion in the Abstract seems to be the most pertinent: "...the study intends to improve palliative care integration for seriously ill older adults in the ED and intensive care unit." The authors clearly understand the need for Goals of Care conversation and the role of Palliative Care within the ED. The introduction of time limited trials in the ED is challenging, as the clinical trajectory could be unclear early in the patient's presentation and the goals that are discussed and set forth may not align with the ICU providers' who will be responsible in carrying them out. If the main points that are to be discussed are overall patient wishes, this aligns more with the framework of a general goals of care conversation than a TLT. The authors need to delineate more specifically the exact points of discussion are to be and how these conversations are separate form a more rudimentary goals of care conversation.

Reviewer #2: My major concern

The authors indicated, "No previous two-arm trial of TLT have been conducted; thus, the effect size remains unknown. Given the primary purpose of conducting the RCT is to demonstrate the feasibility, no gold standard for sample size calculations exists. However, prior literature suggests that 15 to 20 participants per group are required to ensure the scientific validity of the pilot study results [45].” The authors then decided to select 40 participants with 20 each intervention and control group. The authors then went on to say that “In addition, the results of this pilot study will provide information on the difference between groups and the standard deviation necessary for sample size calculation during the future full-scale study.”

These statements are contradictory. Now my question is if the study is not powered enough to see the minimum detectable effect (MID), on what basis can we use the MID estimate from a pilot study to power future studies. I agree that pilot studies have been used to estimate effect sizes to determine the sample size needed for a larger-scale randomized controlled trial (RCT) or observational study. However, this practice has been challenged because pilot study samples are usually small and unrepresentative, as this study has demonstrated by interviewing a total of 40 clinicians (20 controls and 20 interventions). There was no power analyses to arrive at those numbers which means that estimates of parameters and their standard errors may be inaccurate, resulting in misleading power calculations for even larger-scale studies. So adding that the “ the results of this pilot study will provide information on the difference between groups and the standard deviation necessary for sample size calculation during the future full-scale study.” May be problematic. Because the focus is on feasibility, results of statistical tests are generally not informative for powering the main trial outcomes. Additionally, feasibility process outcomes may be poorly estimated because the study is not powered.

The study is not powered since there was no power analysis and authors decided to only do 40 observations, so on what basis will the authors employ all these rigorous statistical analyses (two-sample t-test, binomial test of proportion, Wilcoxon rank sum test) and even run regression models. Remember the standard error estimation or precision of point estimate is based on the sample size which has effect on the estimation of p-values. So if no power analysis was done, how would you standard errors, confidence interval and p-values be reliable?

Also, note that small sample sizes can result in an imbalance between arms or within subgroups that cannot be detected with pilot data or early on in studies. Most pilot studies with smaller sample size should not be used to estimate effect sizes, provide power calculations for statistical tests or perform exploratory analyses of intervention impact.

My suggestion: I think the authors should focus on the primary objective of conducting feasibility studies which is to assess the feasibility/acceptability of Integrating the ‘Time-limited Trial’ in the Emergency Department (the time it takes to conduct the TLTs by ED clinicians) and clinician and patient-reported acceptability of the TLT but without a more rigorous power analysis, the statistical analyses at these stage should be more descriptive in personal opinion

OR if the intervention is non-evasive, you can assume MDE of say 10 percentage point or whatever based on expert advice and what you anticipate the impact of the intervention would be and go through the rigorous power analyses calculation (type I error, power of the study, design effect etc). In that case, all these inferential models can be justified

See the following references.

Teresi, J. A., Yu, X., Stewart, A. L., & Hays, R. D. (2022). Guidelines for designing and evaluating feasibility pilot studies. Medical care, 60(1), 95-103.

Kraemer HC, Mintz J, Noda A, et al. Caution regarding the use of pilot studies to guide power calculations for study proposals. Arch Gen Psychiatry 2006;63(5):484–489.

There was no specific mention of how the data will be analyzed, which is a crucial aspect that the authors should kindly include. Descriptive statistics may be examined but the specific method should be mentioned based on measurement scale of the indicator. For example, the mean and standard deviation for continuous measures (or median and interquartile range for non-normally distributed outcomes) and the frequency and percentage for categorical measures can be calculated overall and by subgroups (intervention and control) without necessarily reporting inferential statistics (p-values, etc).

7. PLOS authors have the option to publish the peer review history of their article (what does this mean?). If published, this will include your full peer review and any attached files.

Reviewer #1: No

Reviewer #2: No

---

## [Author Response · Author response to Decision Letter 0]

15 Oct 2024

Dear Reviewers,

Thank you for taking the time to review our manuscript. We have carefully addressed all of your comments and suggestions in the detailed Response Letter, which has been uploaded for your reference. We appreciate your constructive feedback and hope our revisions meet your expectations.

Sincerely,

Tadayuki Hashimoto

---

## [Editor Report · Decision Letter 1]

1 Nov 2024

Study Protocol for a Randomized Controlled Trial: Integrating the ‘Time-limited Trial’ in the Emergency Department

PONE-D-24-11345R1

Dear Dr. Hashimoto,

We’re pleased to inform you that your manuscript has been judged scientifically suitable for publication and will be formally accepted for publication once it meets all outstanding technical requirements.

Kind regards,

Dong Wook Chang

Academic Editor

PLOS ONE

---

## [Editor Report · Acceptance letter]

6 Dec 2024

PONE-D-24-11345R1 

PLOS ONE

Dear Dr. Hashimoto, 

I'm pleased to inform you that your manuscript has been deemed suitable for publication in PLOS ONE. Congratulations! Your manuscript is now being handed over to our production team.

Kind regards, 

on behalf of

Dr. Dong Wook Chang 

Academic Editor

PLOS ONE